# Development of Biologically Active Phytosynthesized Silver Nanoparticles Using *Marrubium vulgare* L. Extracts: Applications and Cytotoxicity Studies

**DOI:** 10.3390/nano14100895

**Published:** 2024-05-20

**Authors:** Alina Ioana Lupuliasa, Răzvan Mihai Prisada, Roxana Ioana Matei (Brazdis), Sorin Marius Avramescu, Bogdan Ștefan Vasile, Radu Claudiu Fierascu, Irina Fierascu, Bianca Voicu-Bălașea, Marina Meleșcanu Imre, Silviu-Mirel Pițuru, Valentina Anuța, Cristina Elena Dinu-Pîrvu

**Affiliations:** 1Department of Physical and Colloidal Chemistry, Faculty of Pharmacy, “Carol Davila” University of Medicine and Pharmacy, 6 Traian Vuia Str., 020956 Bucharest, Romania; alina-ioana.lupuliasa@drd.umfcd.ro (A.I.L.); valentina.anuta@umfcd.ro (V.A.); cristina.dinu@umfcd.ro (C.E.D.-P.); 2Innovative Therapeutic Structures Research and Development Centre (InnoTher), “Carol Davila” University of Medicine and Pharmacy, 6 Traian Vuia Str., 020956 Bucharest, Romania; 3National Institute for Research & Development in Chemistry and Petrochemistry—ICECHIM Bucharest, 202 Splaiul Independenței, 060021 Bucharest, Romania; roxana.brazdis@icechim.ro (R.I.M.); irina.fierascu@icechim.ro (I.F.); 4Faculty of Chemical Engineering and Biotechnology, National University of Science and Technology Politehnica Bucharest, 1-7 Gheorghe Polizu St., 011061 Bucharest, Romania; 5Department of Inorganic Chemistry, Organic Chemistry, Biochemistry and Catalysis, Faculty of Chemistry, University of Bucharest, 030018 Bucharest, Romania; sorin.avramescu@g.unibuc.ro; 6Research Centre for Environmental Protection and Waste Management (PROTMED), University of Bucharest, Splaiul Independenței 91-95, Sect. 5, 050107 Bucharest, Romania; 7Research Center for Advanced Materials, Products and Processes, National University of Science and Technology Politehnica Bucharest, 060042 Bucharest, Romania; bogdan.vasile@upb.ro; 8National Research Center for Micro and Nanomaterials, National University of Science and Technology Politehnica Bucharest, 060042 Bucharest, Romania; 9Faculty of Horticulture, University of Agronomic Sciences and Veterinary Medicine of Bucharest, 59 Mărăști Blvd., 011464 Bucharest, Romania; 10Interdisciplinary Centre for Research and Development in Dentistry (CICDS), Faculty of Dental Medicine, “Carol Davila” University of Medicine and Pharmacy, 020021 Bucharest, Romania; bianca.voicu-balasea@drd.umfcd.ro (B.V.-B.); marina.imre@umfcd.ro (M.M.I.); silviu.pituru@umfcd.ro (S.-M.P.); 11Department of Prosthodontics, Faculty of Dentistry, “Carol Davila” University of Medicine and Pharmacy, 17-23 Calea Plevnei, 010221 Bucharest, Romania; 12Department of Organization, Professional Legislation and Management of the Dental Office, Faculty of Dentistry, “Carol Davila” University of Medicine and Pharmacy, 17-23 Plevnei Street, 020021 Bucharest, Romania

**Keywords:** phytosynthesis, *Marrubium vulgare* L., silver nanoparticles, extract characterization, biomedical applications

## Abstract

Metal nanoparticle phytosynthesis has become, in recent decades, one of the most promising alternatives for the development of nanomaterials using “green chemistry” methods. The present work describes, for the first time in the literature, the phytosynthesis of silver nanoparticles (AgNPs) using extracts obtained by two methods using the aerial parts of *Marrubium vulgare* L. The extracts (obtained by classical temperature extraction and microwave-assisted extraction) were characterized in terms of total phenolics content and by HPLC analysis, while the phytosynthesis process was confirmed using X-ray diffraction and transmission electron microscopy, the results suggesting that the classical method led to the obtaining of smaller-dimension AgNPs (average diameter under 15 nm by TEM). In terms of biological properties, the study confirmed that AgNPs as well as the *M. vulgare* crude extracts reduced the viability of human gingival fibroblasts in a concentration- and time-dependent manner, with microwave-assisted extracts having the more pronounced effects. Additionally, the study unveiled that AgNPs transiently increased nitric oxide levels which then decreased over time, thus offering valuable insights into their potential therapeutic use and safety profile.

## 1. Introduction

Over the past few decades, the development of phytosynthesized metallic nanoparticles has become one of the most productive research areas in nanotechnology. The annual publication volume on this topic has been growing exponentially. Having several advantages compared with nanoparticles obtained by classical routes (including, but not limited to, increased antioxidant or antimicrobial properties, as well as reduced toxicity), phytosynthesized nanoparticles are now considered viable alternatives to conventional metal nanoparticles [1]. Extracts from a diverse range of plants are employed in the green synthesis of these nanoparticles, where they act as reducing and capping agents.

Despite the vast array of plants explored for this application, a review of major scientific databases (SCOPUS and Web of Knowledge) revealed no studies on the use of *Marrubium vulgare* L. extracts for the phytosynthesis of metal nanoparticles. Commonly known as horehound, *M. vulgare* is mainly found in Asia and Europe. It is a perennial plant of about 40 cm, and is covered in its young stage by a white cotton-like felt. It can grow in almost any soil type but prefers calcareous soils, in dry and sunny environments. Harvesting of leaves is carried out before complete greening. *M. vulgare* has a musk-like scent that fades upon drying and presents a bitter yet aromatic and pleasant taste [2].

Horehound is a medicinal herb that has been historically used for its expectorant and antispasmodic properties. Traditionally, it is utilized in treating respiratory conditions like coughs, bronchitis, and asthma [3,4]. Consequently, *M. vulgare* is often found in herbal remedies for respiratory health.

Numerous secondary metabolites have been isolated and identified from various parts of the plant, including diterpenes, flavonoids, and sesquiterpenes. Notably, researchers have identified over ten different flavonoids within various parts of *M. vulgare*. Additionally, to date, nine distinct types of diterpenes have been reported, including alcohol derivatives. Due to the presence of these compounds, *M. vulgare* shows potential for various pharmacological uses [5,6,7].

Literature data propose *Marrubium vulgare* L. extracts as potent anti-inflammatory [8,9] and antihypertensive [10] agents, as a source of vasorelaxant secondary metabolites [11], or as antidiabetic and antidyslipidemic agents [12,13,14,15]. Based on its rich phenolic secondary metabolites, the horehound extracts can be successfully used as antioxidant agents [16,17,18], with several studies evaluating its potential through in vitro assays such as the DPPH assay [19,20,21]. Other potential applications of horehound extracts or essential oils are based on their antimicrobial effect. The studies report an antibacterial potential comparable to the standard drug ciprofloxacin [22], effectiveness against several Gram-positive bacterial species (*Staphylococcus aureus* and *S. epidermidis*) [23], as well as activity against pathogens like *Salmonella enterica*, *Listeria monocytogene*, *Pseuodomonas aeruginosa*, *Helicobacter pylori*, *Mycobacterium tuberculosis* and *Agrobacterium tumefaciens* [24,25,26].

On the other hand, in recent decades, nanomaterials have been increasingly applied across various industries and everyday life. They are used in sunscreens for effective UV protection, in cosmetics to improve texture, enhance color, ensure superior product adherence, and control the release of active ingredients in skincare products. Nanomaterials are also utilized in textiles, food packaging, water purification, the electronics industry, medicine (for drug delivery, imaging, and diagnostics), as catalysts in chemical processes, as fuel additives, and in coatings for various applications, such as anti-scratch and anti-reflective coatings on eyeglasses, and self-cleaning surfaces [27,28,29,30]. While nanoparticles provide numerous advantages, there are also concerns about their potential environmental and health impacts. These concerns are continually being studied and addressed by researchers and regulatory bodies to ensure the safe and responsible use of nanoparticles in various applications.

In light of these considerations, this study seeks to combine the potential of nanomaterials with the benefits of horehound extracts by exploring the ability of *M. vulgare* extracts to phytosynthesize silver nanoparticles (AgNPs) and evaluating their biomedical properties. The workflow for the study design is illustrated in Figure 1.

## 2. Materials and Methods

### 2.1. Materials

The vegetal material used for the studies is represented by certified horehound flowering aerial parts, purchased from the local market. Prior to the extraction procedures, the dried vegetal material was uniformly shredded to obtain fragments under 2 mm (as determined by sieving). To obtain the natural extracts, two methods were applied: a classical method (involving the mixing of the vegetal material with the solvent and its extraction using a laboratory oven for three hours at 70 °C) and a microwave-assisted method for which the vegetal material and the solvent were heated using an Ethos Easy Advanced Microwave Digestion System (Milestone Srl, Sorisole, Italy) for 30 min. at a temperature of 70 °C and microwave power 800 W. The extracts were encoded as MT (*Marrubium* temperature) and MM (*Marrubium* microwave), respectively. For both methods, the solvent used was a hydroalcoholic mixture (ethanol: water 1:1 *v*/*v*) and the ratio of vegetal material to solvent was maintained at 1:10 (*w*/*v*). The ethanol used was reagent quality (Chimreactiv, Bucharest, Romania), while the water used for all experiments was bidistilled water obtained in the laboratory (GFL 2102 water still, GFL, Burgwedel, Germany). The obtained extracts, after cooling to room temperature, were filtered using filter paper, and subsequently used for the phytosynthesis procedure. The extracts obtained were divided in two parts (one part further used for the phytosynthesis procedure, and one preserved at 4 ℃ until subjected to specific analyses).

Silver nitrate (AgNO_3_, 10^−3^ M) (Chimreactiv, Bucharest, Romania) aqueous solution was prepared with bi-distilled water. For the phytosynthesis process, equal volumes of extract and silver nitrate solution (25 mL from each solution) were mixed and allowed to react for 24 h, after which the NPs dispersions (encoded MT-AgNP, and MM-AgNP, respectively) were used for further tests.

The reduction of Ag+ was achieved by adding the crude extracts to the silver nitrate freshly prepared solution, the final pH of the mixture (evaluated using pH paper) being around 7. The solutions were kept at room temperature and ambient conditions, without any stirring. The formation of silver nanoparticles can be visually monitored by the apparition of a specific ruby-red color [31,32]. After 24 h, the NPs suspensions were immediately analyzed by UV-Vis spectrophotometry, and the remaining solutions were preserved in the dark at 4 °C and further used for the described tests.

### 2.2. Characterization Methods

The composition of the natural extracts was evaluated by the determination of the total phenolic content and high-precision liquid chromatography (HPLC, for the quantification of several target compounds). All reagents were used as received, without further purification.

The total phenolics content was determined using a colorimetric method (Folin-Ciocâlteu reagent method), as previously described [31,32], involving the formation of a blue complex upon the reduction of the Folin–Ciocâlteu reagent by phenolics present in the extracts. The Folin–Ciocâlteu reagent (Merck KGaA, Darmstadt, Germany) and sodium carbonate solution (Merck KGaA, Darmstadt, Germany) were commercially available and used without any purification. The optical density was determined at 765 nm using a Rigol Ultra 3660 UV-Vis spectrophotometer (Rigol Technologies, Beijing, China). The determined values were compared to a standard curve prepared with gallic acid solutions, and the final results were expressed as milligrams of gallic acid equivalents (GAE)/gram of dried matter [31,32].

Five determinations were performed, and the results were presented as the average of the determinations ± the standard error of the mean.

The quantification of polyphenols and the other compounds present in the extracts was carried out using an L-3000 HPLC system (Rigol Technologies Inc., Beijing, China) equipped with a diode-array detector (HPLC-DAD) and a Kinetex EVO C18 column, 150 × 4.6 mm, particle size of 5 µm (Phenomenex, Torrance, CA, USA). The mobile phase consisted of a system with two solvents, and the elution was carried out in gradient mode. The solvents used were (A) 0.1% trifluoroacetic acid (TFA) in water and (B) 0.1% TFA in acetonitrile. The elution gradient was as follows: 2–100% solvent (B) at 30 °C for 60 min at an elution flow rate of 1 mL/min. The analysis was performed at 5 different wavelengths (255, 280, 325, and 355 nm) in accordance with the specialized literature. The stock solutions containing the reference compounds (belonging to several classes: phenolic acids—gallic acid, protocatechuic acid; flavonoids—isoquercitrin, myricetin, catechin, epicatechin, hyperoside, naringin, naringenin, luteolin; benzoic acid derivatives—vanillic acid; hydroxycinnamic acids and derivatives—caffeic acid, sinapic acid, o-coumaric acid, p-coumaric acid; tannins and derivatives—tannic acid, ellagic acid; chlorogenic acids—chlorogenic acid; phytoalexins—resveratrol; all standards Merck KGaA, Darmstadt, Germany) were prepared so that their concentration was 1000 µg/mL. For the calibration curves, concentrations between 10 and 400 µg/mL were used.

The formation of silver nanoparticles under the action of horehound extracts was first monitored using UV-Vis spectrophotometry. Silver nanoparticles usually exhibit absorption maxima in the 400–550 nm region, the exact position of the specific peak being size-dependent [31,32]. For the determinations, a Rigol Ultra 3660 (Rigol Technologies Inc., Beijing, China, optical resolution 0.5 nm, 1 cm path length quartz cuvette) device was used, with a wavelength range of 370–600 nm. Prior to analysis, the nanoparticle solutions were diluted using bidistilled water (dilution factor 100), and correspondingly, diluted extracts were used for comparison purposes. The registered adsorption spectra correspond to a silver concentration of 0.5 × 10^−5^ M.

In order to evaluate the general composition of the extracts, crude undiluted extracts were analyzed in the same wavelength interval.

X-ray diffraction (XRD) analyses were performed with a 9 kW Rigaku SmartLab diffractometer (Rigaku Corp., Tokyo, Japan, operated at 45 kV and 200 mA, CuKα radiation—1.54059 Å), in scanning mode 2θ/θ, between 7 and 90° (2θ). The separation of peaks and phase identification were performed using PDXL 2.7.2.0. software (Rigaku Corporation, Tokyo, Japan). Components were identified by comparison with ICDD data. NP solutions were centrifuged using a DLAB DM0408 (DLab, Beijing, China) laboratory centrifuge at 4000 rpm for three hours. After removing the supernatant, the samples were deposited on specific glass supports for analysis. A similar approach was applied for the XRD analysis of crude extracts (MT and MM), for comparison purposes.

Transmission Electron Microscopy (TEM) analysis was performed using a Titan Themis 200 image corrected transmission electron microscope (FEI, Hillsboro, OR, USA), equipped with a high-brightness field emission gun (X-FEG) electron source and a Super-X detector for energy dispersive spectroscopy (EDX). The heterostructures were investigated at 200 kV by HR-TEM (high-resolution TEM), coupled with selected area electron diffraction (SAED) used for structural identification.

In order to evaluate the phytoconstituents involved in the phytosynthesis procedure, FT-IR (Fourier-transform infrared spectroscopy) analyses were performed both on the crude extracts and on the separated nanoparticles (using the same procedure as described for XRD analysis). FTIR measurements were performed with a JASCO FT-IR 6300 instrument (Jasco Int. Co., Ltd., Tokyo, Japan), equipped with a Specac ATR Golden Gate (Specac Ltd., Orpington, UK) with KRS5 lens, in the range of 400 to 4000 cm^−1^ (32 accumulations at a resolution of 4 cm^−1^), the samples being directly placed on the ATR crystal.

### 2.3. Cell Culture

Human fibroblasts, gingiva (HFIB-G) cell line, were purchased from Provitro (Berlin, Germany). The cells were grown in a humidified atmosphere at 37 °C with 5% CO_2_ in complete DMEM supplemented with 10% fetal bovine serum and 1% penicillin/streptomycin/amphotericin. For the experiment, with a density of 10^4^ fibroblasts per well, the fibroblasts were seeded in 96-well plates for 24 h. After that, the cells were incubated with different concentrations (0.1%, 2.5%, 5% dilutions in complete Dulbecco’s modified Eagle’s medium of the MM and MT extract solutions, as well as their derived nanoparticles, MM-AgNP and MT-AgNP) for the next 24 and 48 h. Control samples were represented by cells unexposed to phytosynthesized NPs.

### 2.4. Cell Viability Assay

The 3-(4,5-dimethylthiazol-2-yl)-2,5-diphenyltetrazolium bromide (MTT, Thermo Fisher Scientific, Eugene, OR, USA) assay was used to determine viability of the cells after 24 and 48 h of cell growth in the presence ofcrude extracts and phytosynthesized NPs, as directed by the manufacturer. A FLUOstar^®^ Omega multi-mode microplate reader from BMG LABTECH (Ortenberg, Germany) was used to measure absorbance at 570 nm after 4 h of incubation with MTT and another 4 h of incubation of SDS–HCl solution (10%) at 37 °C. The SDS–HCl solution (10%) was used to dissolve the purple formazan crystals generated in the viable cells. The results obtained were compared to the control.

### 2.5. Griess Assay

The level of inflammation is suggested by NO levels accumulated in the cell growth medium. A Nitric Oxide Assay Kit (NO, Thermo Fisher Scientific, Vienna, Austria) based on the Griess reagent was used to measure the NO level after 24 and 48 h of incubation. The absorbance at 540 nm was measured using a BMG LABTECH’ FLUOstar^®^ Omega multi-mode microplate reader (BMG LABTECH, Ortenberg, Germany), and the results were compared to the control.

### 2.6. Statistical Interpretation and Data Representation

Determinations were carried out by multiple parallel determinations (as mentioned for each method), and the data obtained were analyzed for statistical significance using analysis of variance (one-way ANOVA) and Tukey’s test to determine significant differences between means. Significant differences were set at *p* ≤ 0.05. The results shown are mean ± standard error of the mean (SE) of independent determinations.

Graphical representations were constructed using the OriginPro 2018 Data Analysis and Graphing Software v9.5.1 (OriginLab Corporation, Northampton, MA, USA).

## 3. Results

Following the extraction, analytical assays were performed to obtain an image of their composition. The results are presented in Table 1.

UV-Vis confirmation of the nanoparticles formation is presented in Figure 2.

For the evaluation (Figure 2a,b), diluted samples were used (for both the nanoparticle solutions and extracts, dilution factor = 100). The UV-Vis adsorption spectra confirm the phytosynthesis of silver nanoparticles, with specific AgNP peaks recorded around 423 nm (sample MT-AgNP) and 430 nm (sample MM-AgNP), suggesting slightly higher dimensions for the MM-AgNPs; however, due to the qualitative nature of this estimation, this should be further confirmed by the other analytical methods.

The results of the XRD analysis of the nanoparticle dispersions are presented in Figure 3.

In order to evaluate the dimensions and morphologies of the obtained nanoparticles, TEM analysis was performed by placing 10 µL of diluted (1:10, *v*/*v*) and ultrasonicated NP sample on a 400-mesh lacy carbon-coated TEM copper grid. The TEM images and particle size distribution of the nanoparticles are presented in Figure 4.

The evaluation of both samples via TEM and HR-TEM SAED sand particle size distributions demonstrated that the particles are in general spherical, with an average diameter of 14.4 nm (for sample MT-AgNP) and 18.8 nm (for sample MM-AgNP).

Figure 5 presents the FTIR spectra recorded for the phytosynthesized NPs, compared with the used extracts.

The result for the MTT assay showed that both crude extracts and their corresponding nanoparticles demonstrate a concentration-dependent decrease in cell viability, with microwave-assisted extracts (MM) and microwave-synthesized nanoparticles (MM-AgNP) exhibiting more pronounced effects than their temperature-extracted counterparts (Figure 6a,b). Specifically, MM-AgNPs reduced HFIB-G cell viability by 37.72% after 24 h and by 56.85% after 48 h at a 5% concentration (Figure 6a), whereas for MT-AgNPs, the reduction was of 18.08% after 24 h and 38.86% after 48 h (Figure 6b).

Comparative analysis of the cytotoxic effects between the crude extracts and their respective nanoparticles revealed a consistent pattern, whereby crude extracts displayed greater potency. Notably, the MM extract resulted in a substantial 69% reduction in cell viability after 48 h, exceeding the reduction observed for its nanoparticle counterpart (56.86%) (Figure 6a). This trend underlines the enhanced cytotoxic capabilities of crude extracts compared to their nanoparticle formulations, suggesting a potential concentration of bioactive compounds in the extracts that may be moderated when processed into nanoparticles.

Nitric oxide (NO) levels, measured using the Griess assay, demonstrated an initial increase after 24 h of incubation with both MM-AgNP and MT-AgNP as compared to the control (Figure 7a,b). However, a decrease in NO levels was observed after 48 h, suggesting a transient inflammatory response (Figure 7a,b). In contrast, when using crude extracts, NO levels increased for both extraction methods and remained elevated after both 24 and 48 h at the highest concentration (5% of the initial extract), indicating a sustained inflammatory response compared to the control. Notably, the most significant increase in NO, by 27%, was observed after 48 h of incubation with the MM extract, aligning with the cytotoxicity findings reported in the MTT assay results.

## 4. Discussion

The results regarding extract composition are in good agreement with available literature data. Abidi et al. [33] presented a total phenolic content of over 73 mg GAE/g of extract for an aqueous extract obtained by hot infusion. At the same time, the cited study identified the presence of several metabolites quantified in the present study (gallic acid, protocatechuic acid, catechin, chlorogenic acid, caffeic acid, epicatechin, rutin). Amri et al. [34] obtained a total phenolic content in a hydroalcoholic extract (80% methanol) obtained by a microwave-assisted method of approx. 6 mg GAE/g dry weight, slightly lower than our result, referring to an extract obtained by a comparable method (sample MM).

Mssillou et al. [35] obtained, for horehound extract obtained by maceration using 70% ethanol from Moroccan plants, a total phenolic content significantly higher than the present study; significantly higher levels of TPC were also recorded by Amessis-Ouchemoukh [36] for Algerian native horehound. As also confirmed by Bouterfas et al. [37] who observed high variation in the TPC for plants with different origin, the differences can be attributed to several factors, including differences in genotype, environment, plant parts used, solvent or extraction method [35]. In our case, a statistically significant difference is recorded in terms of TPC between samples MM and MT, the microwave-assisted method leading to the extraction of superior amounts of total phenolics. This is also supported by the HPLC results; in most of the cases, the MM sample had significantly higher contents of the quantified compounds. Several of these target compounds (caffeic acid and luteolin, and gallic acid) were also identified by Amessis-Ouchemoukh [36] and Al-Zaban et al. [38], respectively.

Regarding the registered levels of target compounds, there are some studies presenting the HPLC analysis of horehound extracts, which can be used as literature reference data. For example, Rezgui et al. [39] identified and quantified a methanolic extract obtained at room temperature (in g/100 g dry weight): p-cumaric acid—0.00093, caffeic acid—0.0024 and luteolin—0.066; Wojdylo et al. [40] revealed the presence of (in mg/100 g dry weight) caffeic acid—166 and p-coumaric acid—31.6; while Kabach et al. [41] reported several compounds in an aqueous extract (in μg/g dry weight): caffeic acid–0.258, gallic acid—0.363, p-coumaric acid—0.072 and naringenin—0.145. Another observation made by Kabach et al. [41] was that p-coumaric acid and naringin were found in significantly higher amounts in the methanolic extract (1.015 and 1.567 μg/g dry weight, respectively), while the other compounds were not recorded (proposing the methanolic extraction as a more selective route). Considering the extraction method used in the present study, the results obtained are comparable with the literature data.

A particular observation regarding the HPLC results is that the only two compounds that are found in higher quantities in the MT sample (compared with MM) are flavan-3-ol catechin, and, in a lesser extent, the hydroxycinnamic acid o-coumaric acid (it is noteworthy that the other isomers quantified, epicatechin and p-coumaric acid, are found in larger quantities in MM). These observations would suggest some selectivity of the extraction methods that could influence the formation of nanoparticles and biomedical applications.

Regarding the analytical evaluation of the NP formation, it can be noticed that all three methods used for their evaluation (UV-Vis spectrophotometry, XRD, TEM and FTIR) lead to a similar conclusion, that the MT leads to smaller-dimension NPs compared with the extract MM. A similar observation was made by Fierascu et al. [31] regarding the use of *Echinacea* extracts for the phytosynthesis of silver nanoparticles, although in that particular case, the total phenolic content was also higher in the extract obtained by classical temperature extraction. Considering the obtained results, the most likely observation is that other groups of secondary metabolites (i.e., terpenoids, or alkaloids) play a major role in the phytosynthesis of silver nanoparticles using the horehound extracts. As such, the influence of the polyphenols (extracted in higher quantities in sample MM) is counter-balanced by other classes (not quantified in the present study).

The UV-Vis spectra for samples MT-AgNP and MM-AgNP (Figure 2a,b) display absorbance peaks in the 400–500 nm range, indicative of nanoparticle phytosynthesis. As noted earlier, even though the UV-Vis absorption spectra are recorded using diluted samples, they not only confirm the phytosynthesis process but also suggest slightly larger nanoparticle sizes for the MM-AgNP sample. When compared to the absorption spectra of similarly diluted extracts and a silver nitrate stock solution (Figure 2a,b), neither the extracts nor the silver nitrate show any absorption peaks in the relevant region.

A general evaluation of the extracts’ composition can also be performed using the UV-Vis absorption spectra (Figure 2). As literature data suggest [42,43,44,45], the strong adsorption bands up to 450 nm are mainly due to the presence of phenolic compounds (the exact location of the maxima being affected by several factors, including conjugation degree, number and position of substituents, etc.); another band appears in the spectra recorded for both extracts (around 660 nm), which is usually assigned to the presence of chlorophylls [45].

The phytosynthesis process is confirmed by the XRD analysis (Figure 3), in which, particularly, the peak corresponding to the (111) plane is clearly visible. The identification of the Ag^0^ phase was confirmed by comparison with the ICDD PDF card no. 01-077-6577. The presence of the other XRD peaks, although slightly visible due to the contribution of the amorphous halo to the general appearance of the diffractograms, was confirmed by a detailed analysis performed using the PDXL 2.7.2.0. software (Rigaku Corporation, Tokyo, Japan). However, due to the low signal-to-noise ratio, their characteristics were not used for any determinations. Additionally, the crystallite size could not be reliably determined from the data due to the limitations previously described, along with significant methodological errors referenced in sources [46,47]. Furthermore, the X-ray diffractograms of the crude extracts (also depicted in Figure 3) demonstrate that these extracts do not contribute to the formation of the XRD peak assigned to (111) silver lattice plane, as only the specific amorphous halo can be observed [48].

A more accurate estimation of the NP diameters was obtained by TEM. As confirmed by the TEM analysis, the extract MM leads not only to higher dimensions NPs, but also to a wider range of morphologies. Although quasi-spherical NP are the most common, several other types of morphologies can be noticed for sample MM-AgNP: triangular, hexagonal, ellipsoidal, etc. This aspect also supports the use of MT to obtain not only smaller-dimension NP, but also with a more uniform morphology, an important aspect for biomedical applications.

According to SAED images, we can identify cubic silver (ICDD card no. 01-077-6577). HR-TEM images present highly crystalline particles, in which we can identify (111) orientation for both cases. Also, the images presented in Figure 4 demonstrate the fact that the particles are polycrystalline in nature. Another interesting aspect revealed by the TEM analysis is a low-contrast area that is also visible, which can be associated with organic components of the extract, thus further supporting the claim that the phytoconstituents act as capping agents in the phytosynthesis procedure.

In order to evaluate the phytoconstituents involved in the process, FTIR analysis was performed on the NPs and on the parent extracts. Both the extracts and the NPs revealed the presence of functional group corresponding to alcohol and phenols (O–H), carboxylic acids (C–O stretching), methyl and aldehyde group (stretching of C–H bonds), alkenes (C=C stretching), and aromatics (C–C stretching).

The FTIR peaks observed in the two extracts can be assigned as follows: 3383/3365 cm^−1^—free hydroxyl groups; 2979 and 2902/2903 cm^−1^—C–H stretching vibrations; 1652 cm^−1^—carbonyl group vibrations; the bands at 880 and 1453 (only present in MM) could be assigned to the stretching vibrations of CO_3_^2−^; 1382/1384 to 1044/1045 cm^−1^—asymmetrical and symmetrical stretching vibrations of PO^−2^ and phospholipids; 1404 cm^−1^—stretching of C–O–C of nucleic acids and phospholipids; 632 cm^−1^—out-of-plane bending of CH vibrations; the bands in the region 700–900 cm^−1^ correspond to out-of-plane bending vibrations; and the bands in the region 500–600 cm^−1^ were attributed to ring stretching vibrations strongly mixed with in-plane bending of CH vibrations. The results are in good concordance with available literature data [49]. When comparing the results obtained for the NPs with the extracts, an increase in the peaks corresponding to the organic compounds (1400–1650 cm^−1^) can be noticed. In particular, several new peaks appear in the NP FTIR spectra, while others show a significant increase in intensity as compared to the crude extracts: 1639/1651 cm^−1^, 1608 cm^−1^ (more visible in MM-AgNP), 1516/1515 and 1450/1453, 1272/1248, and 1172/1173, respectively. Literature data [50] assign the apparition of the FTIR peaks 1172/1173 and 1272/1248 cm^−1^ to the presence of AgNPs. The increase observed for the peak at 3363/3331 cm^−1^ is most probably due to the presence of silver nanoparticles, as in [48]. The remaining significant changes appearing in the FTIR spectra could be assigned as follows: 1639/1651 cm^−1^ (C–N and C–C stretching indicating the presence of proteins), 1608 cm^−1^ (more visible in MM-AgNP, corresponding to the amide I band of protein), 1516/1515 (in-plane CH bending vibration from the phenyl rings) and 1450/1453 (N–H stretch vibration present in the amide linkages of the proteins) [51,52]. The results suggest a synergistic action of multiple classes of phytoconstituents in the phytosynthesis process, including phenolic compounds and proteins.

The toxic effects of silver ions are relatively rare in humans. However, chronic exposure triggers the formation of eye (ocular argirosis) and skin (argyria) silver deposits [39]. Regarding the oral mucosa, silver amalgam dental restorations induce the appearance of a blue-black coloration into the surrounding tissue. However, it has been reported that these toxic effects are reversible after stopping exposure.

Given their recent emergence in medicine, the potentially toxic effects of nanomaterials on human health have not yet been thoroughly investigated. Current literature is sparse, particularly concerning the toxicological effects of silver nanoparticles (AgNPs) on various organs, tissues, and cell types. A pivotal in vitro study by Hernández-Sierra et al. [53] highlighted the dose- and time-dependent cytotoxicity of AgNPs smaller than 20 nm on human periodontal fibroblasts, in contrast to larger particles (80–100 nm), which exhibited negligible impact on cell viability. Furthermore, it has been demonstrated that AgNPs can induce osteogenic differentiation in human periodontal ligament fibroblasts via the RhoA-TAZ signaling pathway [54].

To our knowledge, there are no studies presenting the effects of AgNPs phytosynthesized using *M. vulgare* extracts. One of the aims of this study was to evaluate the biocompatibility of the nanoparticles obtained by applying classical temperature extraction (MT-AgNP) and microwave-assisted extraction (MM-AgNP) to human gingival fibroblasts (HGFs). The cytotoxic effects of the two types of AgNPs were evaluated in relation to the HGF viability decrease (MTT test) and the increase in NO levels.

The cell viability experiments revealed that the crude extracts of *M. vulgare* (MM and MT) as well as their corresponding nanoparticles (MM-AgNP and MT-AgNP) exhibit concentration-dependent cytotoxicity towards human gingival fibroblasts, with microwave-assisted methods producing stronger effects than their temperature-based extraction counterparts (Figure 6b). MM-AgNPs exhibited a pronounced cytotoxic effect on HFIB-G cells, with a notable cell viability reduction of 37.72% after 24 h and 56.85% after 48 h at a 5% concentration, substantially higher than MT-AgNPs, which showed an 18.08% reduction at the same concentration after 24 h, escalating with extended exposure up to 38.86% at 48 h. The calculated IC50 value (concentration at which the viability of the tested cells is reduced by 50%) for MM-AgNP at 48 h was approximately 2.7 μg/mL (5% dilution of the initial silver concentration, as can be observed from Figure 6b). Those differences are directly correlated with the *M. vulgare* extraction methodology and align with the findings from the nanoparticle exposures, indicating that the microwave-assisted method potentially disrupts plant cell matrices more effectively and extracts a higher concentration of phytochemicals, thereby amplifying the cytotoxic impact on human gingival fibroblasts of both MM crude extracts and its phytosynthesized nanoparticles.

The comparative analysis of cytotoxic effects between the crude extracts and their respective nanoparticles indicated a consistent trend for the crude extracts to be more potent. The MM extract showed a more substantial decrease in cell viability (69% reduction at 48 h) compared to its nanoparticle counterpart (56.86% reduction) (Figure 6a). This reduction in cytotoxicity compared to the crude extract might be attributed to the encapsulation or binding of bioactive compounds in the nanoparticle synthesis process, which could moderate their bioavailability and interaction with cellular components.

The data from the Griess assay, which measures nitric oxide (NO) levels, provided insightful observations regarding the inflammatory response of human gingival fibroblasts to *M. vulgare*-derived nanoparticles (MM-AgNP and MT-AgNP) and crude extracts. The initial increase in NO levels after 24 h of incubation with both types of nanoparticles, followed by a decrease after 48 h, contrasts with the sustained elevation seen with crude extracts. This pattern reflects distinct dynamics of cellular responses induced by nanoparticles versus crude extracts and offers an opportunity to understand the mechanistic differences in how these substances interact with cellular pathways. The initial increase in NO levels suggests an acute activation of inflammatory pathways. The increase in NO following exposure to nanoparticles could indicate that the cells initially recognize these particles as foreign or potentially harmful, triggering an immune response aimed at neutralizing this perceived threat. This is a typical response where NO acts as a part of the body’s first line of defense, mediating various signaling pathways that activate immune cells and promote inflammatory reactions.

Interestingly, NO levels decreased after 48 h when cells were incubated with nanoparticles, suggesting possible adaptive cellular responses. Given the established connections between NO levels and inflammatory and apoptotic processes, our data might suggest a possible anti-inflammatory effect of the MM-AgNPs, able to evolve in a time-dependent manner.

In contrast to nanoparticles, crude extracts induced a persistent increase in NO levels, which did not subside over time. The MM extract led to a more substantial increase after 48 h, reaching 27% higher NO levels compared to the control, aligning with its higher cytotoxicity. This sustained increase in NO levels is likely attributed to the complex mixture of bioactive compounds in the extracts, which may continuously stimulate the cells by repeatedly triggering immune receptors or inducing ongoing cellular stress, leading to sustained inflammatory signaling.

Eltahawy et al. reported that in their study, a *Marrubium alysson* L. extract was used for the first time to synthesize AgNPs. The authors used these NPs to explore their possible anti-cancer effects on the PC-3 and HCT-116 cell lines. Their results illustrated that the *Marrubium alysson* L.-AgNPs showed apoptosis-inducing capabilities [55]. However, according to our knowledge, there are no reports of biocompatibility and possible anti-inflammatory effects of *M. vulgare*-AgNPs.

Finally, our results suggested that MM-AgNPs and MT-AgNPs, in direct contact with HGFs, induced cytotoxic effects depending on the tested concentrations and exposure times. Our data analysis led to the conclusion that, according to the ISO 10993-1:2018 standard [56], the 0.1% MT-AgNPs showed the best biocompatibility after 24 and 48 h exposure, highlighting the temperature extraction process as more recommendable. Our study contributes to the growing body of evidence on the complex interactions between nanomaterials and biological systems, highlighting the need for comprehensive evaluations of their biocompatibility and therapeutic potential.

The use of natural extracts for the development of metallic nanoparticles represents a field of continuous growth. The presented approach contributes not only to the reduction in production and use of hazardous materials (often used in the case of chemical nanoparticle synthesis) but also, as other studies suggest, to a reduction in the nanoparticles’ toxicity, given the use of secondary metabolites as capping agents. On the other hand, considering the well-known biomedical potential of *M. vulgare* (as presented in the Section 1), the developed nanomaterials could have applications in several biomedical areas, which requires further studies.

## 5. Conclusions

For the first time in the literature data, the phytosynthesis of silver nanoparticles using *Marrubium vulgare* L. extracts was achieved, as demonstrated by the analytical results. With the application of two different extraction methods (classical temperature extraction—MT and microwave-assisted extraction—MM), a significant variation was observed in terms of composition of the natural extracts, which further influenced the silver nanoparticle characteristics. According to the obtained results, the use of classical temperature extraction led to smaller dimensions and more uniform NPs, in terms of morphology and average diameter, as recorded by XRD and TEM, with 14.4 nm for the MT-AgNP sample compared with 18.8 nm for the MM-AgNP sample.

In terms of biological properties, the experimental findings confirmed that both crude extracts and their derived nanoparticles decrease cell viability in a concentration-dependent manner, with microwave-assisted extracts showing notably stronger effects. Additionally, the study highlighted a dynamic inflammatory response, where an initial increase in nitric oxide levels was followed by a decrease with nanoparticles, contrasting with the sustained increase observed with crude extracts, thus providing essential insights for their potential therapeutic application and safety evaluation. The best biocompatibility was observed for the nanoparticles phytosynthesized using classical temperature extraction (MT-AgNPs) applied at a 0.1% concentration.

The current study represents a confirmation of the potential application of *M. vulgare* extracts in the nanotechnology area. The approach aligns with the principle of “green chemistry” contributing to the reduction in hazardous by-products and the use of potentially harmful (for the environment and health) reagents; however, further studies are necessary for the establishment of the toxicity of phytosynthesized nanoparticles.

## Figures and Tables

**Figure 1 nanomaterials-14-00895-f001:**
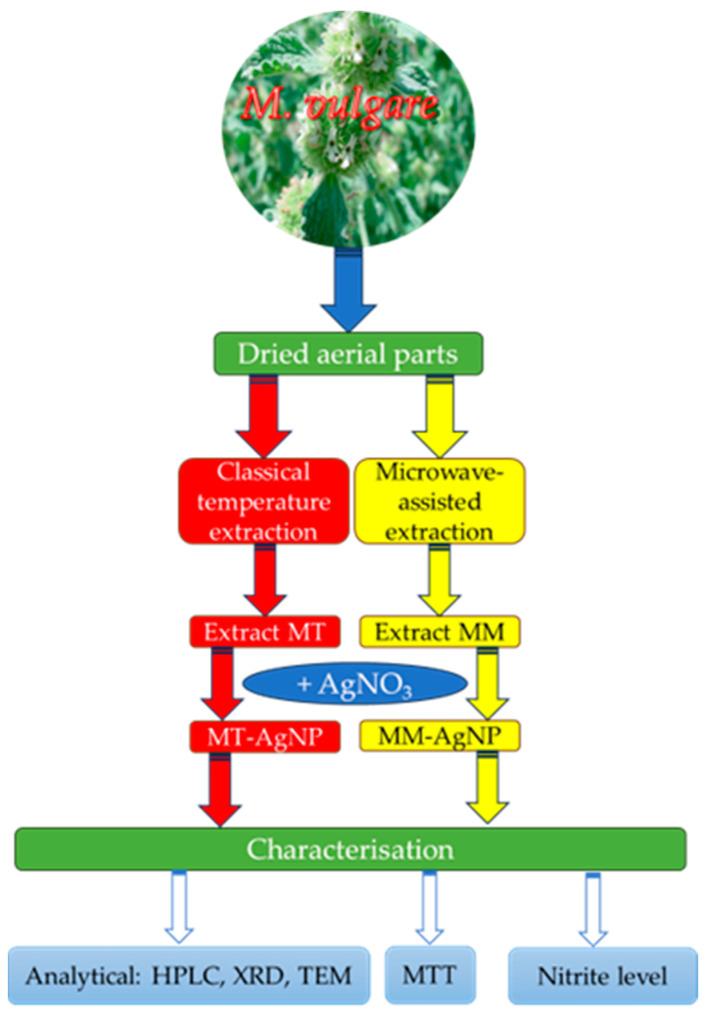
Workflow of the present study.

**Figure 2 nanomaterials-14-00895-f002:**
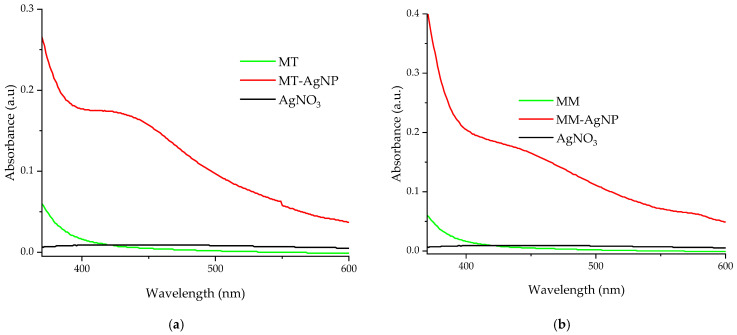
UV-Vis absorption spectra of diluted silver nanoparticles samples, by comparison with the used extracts: (**a**) MT-AgNP; (**b**) MM-AgNP.

**Figure 3 nanomaterials-14-00895-f003:**
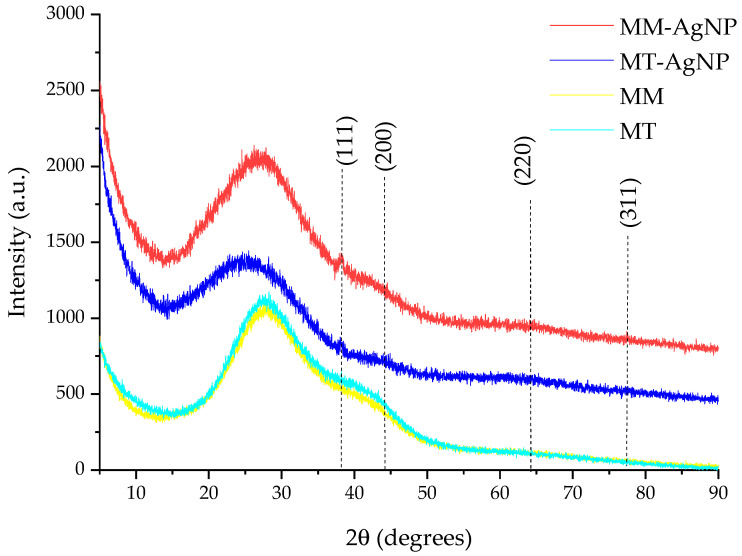
XRD results of the analyzed samples (in the image, the diffraction planes correspond to the identified maxima) and of corresponding extracts.

**Figure 4 nanomaterials-14-00895-f004:**
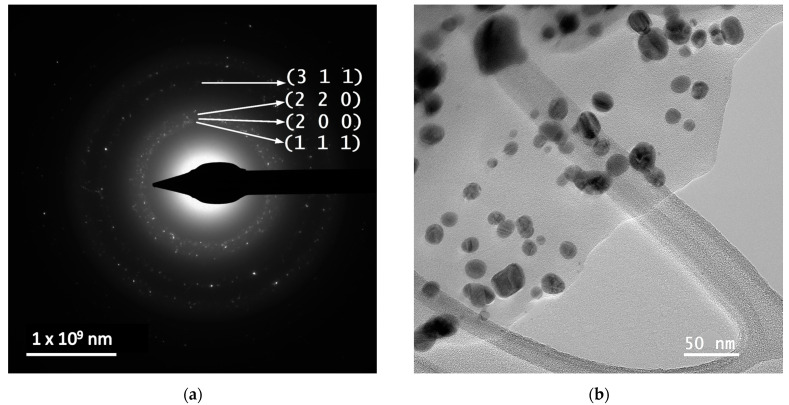
TEM images and particle diameter histogram for samples MT-AgNP—(**a**) SAED image; (**b**) TEM image; (**c**) HR-TEM image; (**d**) particle diameter histogram—and MM-AgNP—(**e**) SAED image; (**f**) TEM image; (**g**) HR-TEM image; (**h**) particle diameter histogram.

**Figure 5 nanomaterials-14-00895-f005:**
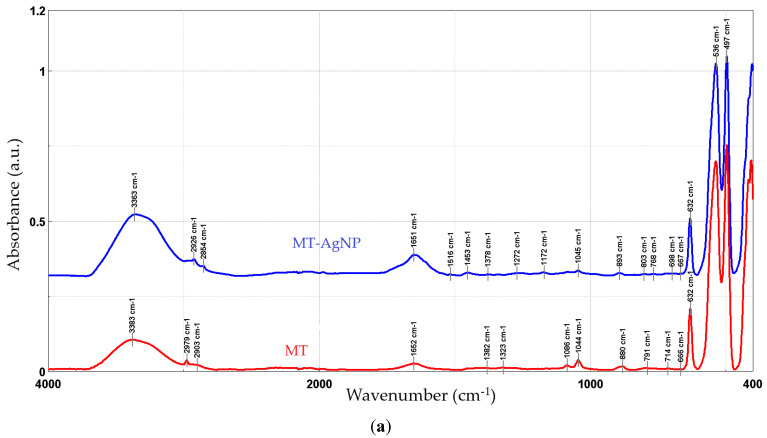
FTIR spectra of the NP samples, compared with the parent extracts: MT-AgNP (**a**) and MM-AgNP (**b**).

**Figure 6 nanomaterials-14-00895-f006:**
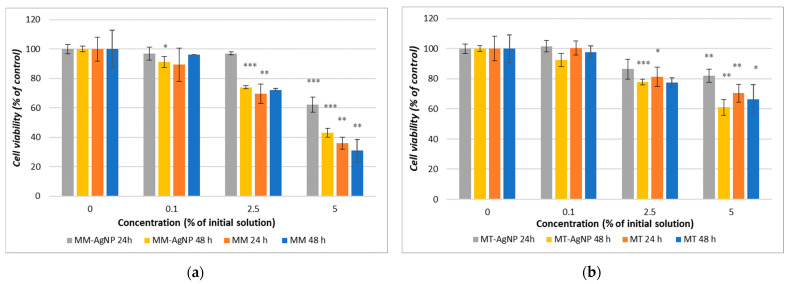
Cell viability after 24 and 48 h exposure of HFIB-G oral cells to different concentrations (expressed as % of initial NP solutions, according to the Section 2) of MM extract and MM-AgNP (**a**) and MT extract and MT-AgNP (**b**). Data are expressed as means ± SD (n = 3). * *p* < 0.05, ** *p* < 0.01, *** *p* < 0.001 compared to control (untreated cells).

**Figure 7 nanomaterials-14-00895-f007:**
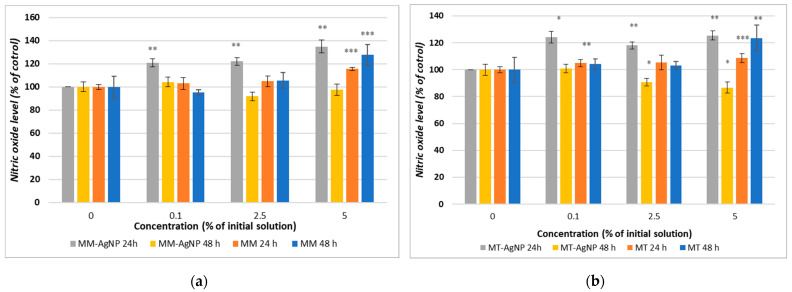
Nitric oxide level measured after the 24 and 48 h incubation of HFIB-G cell line with different concentrations (expressed as % of initial NP solutions, according to Section 2) for MM extract and MM-AgNP (**a**) and MM extract and MT-AgNP (**b**). Data are expressed as the means ± SD (n = 3). * *p* < 0.05, ** *p* < 0.01, *** *p* < 0.001 compared to control (untreated cells).

**Table 1 nanomaterials-14-00895-t001:** Evaluation of extract composition by HPLC and total phenolics content^1^.

Extract/Parameter	MT	MM
TPC (mg GAE/g dried weight)	3.6 ± 0.0179 ^b^	6.47 ± 0.0268 ^a^
Tannic acid (mg/L)	2.7947 ± 0.020766 ^b^	4.5877 ± 0.037933 ^a^
Gallic acid (mg/L)	0.094997 ± 0.00070588 ^b^	0.14717 ± 0.0012168 ^a^
Protocatechuic acid (mg/L)	0.7181 ± 0.0053358 ^b^	1.2731 ± 0.010527 ^a^
Catechin (mg/L)	2.3071 ± 0.017143 ^a^	N.D.
Vanillic acid (mg/L)	0.61427 ± 0.0045643 ^b^	1.3913 ± 0.011504 ^a^
Caffeic acid (mg/L)	2.3912 ± 0.017768 ^b^	4.3165 ± 0.035691 ^a^
Ellagic acid (mg/L)	5.9118 ± 0.043928 ^b^	13.626 ± 0.11267 ^a^
Chlorogenic acid (mg/L)	1.4328 ± 0.010646 ^b^	2.7661 ± 0.022871 ^a^
Epicatechin (mg/L)	9.801 ± 0.072826 ^b^	15.968 ± 0.13203 ^a^
p-coumaric acid (mg/L)	0.09841 ± 0.00073124 ^b^	0.19351 ± 0.0016 ^a^
Sinapic acid (mg/L)	0.54867 ± 0.0040769 ^b^	0.95935 ± 0.0079322 ^a^
o-coumaric acid (mg/L)	0.82797 ± 0.0061522 ^a^	0.64402 ± 0.005325 ^b^
Isoquercitrin (mg/L)	52.031 ± 0.38662 ^b^	98.66 ± 0.81576 ^a^
Hyperoside (mg/L)	21.808 ± 0.16204 ^b^	52.536 ± 0.43439 ^a^
Naringin (mg/L)	0.69081 ± 0.0051331 ^b^	1.2018 ± 0.0099372 ^a^
Myricetin (mg/L)	37.659 ± 0.27983 ^b^	66.758 ± 0.55198 ^a^
Resveratrol (mg/L)	11.25 ± 0.08359 ^b^	19.321 ± 0.15975 ^a^
Luteolin (mg/L)	2.4775 ± 0.018409 ^b^	3.2323 ± 0.026726 ^a^
Naringenin (mg/L)	0.61153 ± 0.004544 ^b^	1.1737 ± 0.0097047 ^a^

Values represent the mean of five determinations for TPC or three determinations for HPLC results ± SE; values in the same row without a common superscript letter differ statistically (*p* < 0.05) as analyzed by one-way ANOVA and the TUKEY test; N.D.—not detected.

## Data Availability

The data presented in this study are available on request from the corresponding authors.

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
