# Peer review of "Development of Biologically Active Phytosynthesized Silver Nanoparticles Using Marrubium vulgare L. Extracts: Applications and Cytotoxicity Studies"

_nanomaterials, 2024, doi:10.3390/nano14100895_

Round 1

Reviewer 1 Report

Comments and Suggestions for Authors

This paper describes the results on the extraction of Marrubium vulgare plant extracts by classical extraction and microwave radiation. The extracts obtained were used to synthesize silver nanoparticles. And the obtained nanomaterials were tested for their cytotoxicity. In general, to date, there are a number of works devoted to the discussion of extraction options from the plants Marrubium vulgare the natural components. The preparation of silver nanoparticles using natural extracts is also intensively investigated. This manuscript contains a number of significant remarks that do not allow us to recommend it for acceptance.

1. The absorption spectra of the plant extracts themselves have not been characterized.

2.Absorption in the region of 580 nm for the spectrum of MM-AgNPs has not been characterized

3. Spherical gold nanoparticles are characterized by the presence of an intense plasmonic absorption maximum in the absorption spectra in the region of 390-440 nm. However, in the spectra presented by the authors, the plasmon resonance band is not highly intense and well resolved. This probably indicates a low percentage of nanoparticles in the sample volume of the analyzed solution, which is also evidenced by the low optical density values. The authors should explain the choice of concentrations to perform absorption spectroscopy and provide extinction spectra for the nanoparticles obtained.

4. The diffractograms of the obtained nanomaterials on which there are no intense reflexes characteristic for Ag0 nanocrystallites cause serious concern. The assignment of reflexes of standard silver with the obtained diffractograms presented by the authors in Figure 2 is assumed. And the value of the coherent scattering area of silver nanoparticles of 2.3 nm calculated by the authors is absolutely not correlated with the data of transmission electron microscopy. There are no maxima in the presented diffractograms.

5. Figure 3 and its reference are missing from the manuscript

6. The dimensions determined by TEM method are absolutely inconsistent with the dimensions determined by XRD. 

7. electronograms presented are not indexed, hence it is not possible to evaluate the phase component.

8. The dimensionality of the scale presented on the SAED data is 1/km and needs to be verified.

In general, this work does not provide reliable information about the obtained nano-objects. All nanomaterials should be characterized in detail by a complex of physicochemical methods of investigation.

Given the high level of the journal "Nanomaterials" and the quality of the material presented in the manuscript to date, I cannot recommend its acceptance. The manuscript should be substantially revised, all nanomaterials should be thoroughly characterized, and their structure should be reliably confirmed. 

Comments on the Quality of English Language

Requires text checking by a native English speaker. The text contains a number of grammatical and stylistic errors. 

Reviewer 2 Report

Comments and Suggestions for Authors

The aim of the manuscript is to present the possibility of synthesis of silver nanoparticles using Marrubium Vulgare L. extracts. The authors characterized the size of the synthesized particles, their optical properties and cytotoxic effects depending on the tested concentrations and exposure times.

I think the manuscript would be interesting because of the novelty of the presented results, but at the same time it is characterized by a number of gaps in terms of the methods used for the analysis of the samples, poor quality in terms of its presentation and incorrectly used terms. Therefore, I cannot recommend that the manuscript be published in its current form.

1. What is the error in determining the crystallite size. Even for larger crystallite sizes than those in the present manuscript, the use of the Debye-Scherrer equation leads to a significant error in the size determination.

[1] P. Scherrer, "Bestimmung der Grösse und der inneren Struktur von Kolloidteilchen mittels Röntgenstrahlen", Nachr. Ges. Wiss. Göttingen 26 98 (1918)

[2] J.I. Langford and A.J.C. Wilson, "Scherrer after Sixty Years: A Survey and Some New Results in the Determination of Crystallite Size", J. Appl. Cryst. 11 102 (1978)

[3] V. Uvarov and I. Popov, "Metrological characterization of X-ray diffraction methods for determination of crystallite size in nano-scale materials", Mater. Charac. 85 111 (2013)

2. The contribution of the amorphous halo in the X-ray patterns in second Figure 2 is significant, and therefore one must imagine how the half-width of the peaks in the X-ray patterns is determined.

3. Confused numbering of figures and their captions 1-4. Figure 3 and its comment are completely missing. TEM images show a significantly larger size of the nanoparticles than the determined size of the crystals.

4. Incorrect use of the term "UV-Vis spectrometry" instead of UV-Vis spectrophotometry

5. Incorrect use of the term UV-Vis spectra. These spectra can be absorption, reflection or transmission.

6. What is the accuracy and how exactly are the positions of the peaks in fig.2 determined and a difference of 7nm since in the case of figure in fig.2b there is no clearly defined absorption peak.

7. What purpose and advantages of using Marrubium Vulgare L. Extracts for phytosynthesized metallic nanoparticles compared to other plant extracts should be added and highlighted in the discussion and conclusion.

Reviewer 3 Report

Comments and Suggestions for Authors

The article by A.I. Lupuliasa with co-authors entitled” Development of Biologically Active Phytosynthesized Silver Nanoparticles using Marrubium Vulgare L. Extracts: Applications and Cytotoxicity Studies” is devoted to the preparation and characterization of Marrubium vulgare L. extract and its use in the preparation of silver nanoparticles. The obtained silver nanoparticles were evaluated in terms of cell viability and anti-inflammatory activity. The use of various plant extracts to reduce silver ions to form nanoparticles is a commonly used method. The authors have done nothing new here. The detailed analysis of the phenolic content depending on the extraction method is interesting but not very informative. The extraction process depends on many factors, including the degree and method of grinding the raw materials. Shredding to 2 mm fragments is not sufficient for characterization. The preparation of silver nanoparticles is also typical. The authors used XRD data to evaluate the crystal size using the Debye-Scherrer equation. However, since the XRD data are of very poor quality (signal-to-noise ratio is too low), it is impossible to obtain reliable data, especially with high precision. According to the TEM images, in addition to the high-contrast nanoparticles, a low-contrast area is also visible, which can be associated with organic components of the extract. In fact, the authors studied the colloidal solution directly after extraction, without separating the AgNPs from the solutions. Furthermore, the authors used these solutions for biological experiments. So, what is the effect of crude horehound extracts on cell viability and oxidative stress? These data should be added. Also, the finding that cytotoxicity is concentration- and time-dependent is not surprising.

In addition to the experiments provided, the authors should determine which and how molecules from the extracts interact with AgNPs and what advantages this provides compared to conventional silver nanoparticles.

In my opinion, this manuscript cannot be published in such a prestigious journal as Nanomaterials in its current form.

Reviewer 4 Report

Comments and Suggestions for Authors

The manuscript ID "nanomaterials-2946841" having the title "Development of Biologically Active Phytosynthesized Silver Nanoparticles using Marrubium Vulgare L. Extracts: Applications and Cytotoxicity Studies" has described study claiming they introduces an innovative green chemistry approach by utilizing Marrubium Vulgare L. extracts for the photosynthesis of silver nanoparticles (AgNPs), highlighting a novel application of this plant in nanomaterials science. Although authors have tried to discuss their results with several approaches, there are some issues with this article. Therefore, it should be revised before publication.

My specific comments on this paper are as follow:

1.      Authors should incorporate a graphical abstract highlighting the importance of the work.

2.      Authors should cite the following essential articles in the introduction section of the manuscript. They can find some related insights in these articles: -

a.       https://doi.org/10.1038/s41598-018-27170-1 .

b.      https://doi.org/10.1002/elan.202300094.

c.       https://onlinelibrary.wiley.com/doi/abs/10.1002/elan.201900337.

3.      Carefully check the abbreviations used in the manuscript. Authors have explained some abbreviations twice, and some are not explained at all. Check for TEM and XRD, MM MT (Explain it in initial part of the manuscript)

4.      The methods section, while detailed, could benefit from more specifics on the preparation and extraction processes to enhance reproducibility by other researchers.

5.      Please add a section of synthesis of Ag NPs, it is very vague to write “allowed to react for 24 hours”, explain in detail about the reaction conditions such as temperature, whether it was stirred or only aged in a flask. What will happen if someone keeps it for 48 hours?

6.      How did authors find that only 24 hours are required for this synthesis? Explain more about standardization of synthesis procedure? Provide some control experiments or observations for synthesis optimization.

7.      XRD is very poor; I would like to see the XRD of plant extract as well (if possible); I am afraid the peaks are coming from plant extract or sample was not prepared well. Try to see the XRD of chemically synthesized Ag NPs and compare it with MT and MM Ag NPs)

8.      Please elaborate more about sample preparation steps for each characterization.

9.      Elemental Line profiling: I didn’t see anything related to this in TEM explanations.

10.  Some figures (e.g., TEM images and particle size distribution histograms) are mentioned but without clear discussion in the text about their implications, leaving readers to infer the significance of these results.

11.  Figure captions are scarce and not detailed; please label every figure with figure numbers.

12.  Authors must indicate crystal planes in SAED analysis; and try to correlate it with XRD analysis.

13.  I am seeing a cover of translucent film in TEM around nanoparticles. It could be flavonoids and phenolic acids from plant extract. It could benefit the claim of the authors if they can show some chemical groups adhered to the Ag NPs coming from plant extract. If possible, do the FTIR of Phyto fabricated AgNPs and chemically synthesized AgNPs with control of plant extract (control).

14.  Please explain more about LD 50 (lethal dose of 50%) for cytotoxicity and oxidative stress.

15.  The discussion on the environmental and health safety aspects of the synthesized nanoparticles is brief. Given the increasing concern over nanoparticle toxicity, a more detailed assessment is necessary.

16.  I have seen many syntax errors; there are errors in punctuation marks such as comma in the whole manuscript. Check for superscript and subscript errors in the whole manuscript. Some sentences are incomplete, making it hard to understand what do authors want to say? Authors should read and revise the manuscript for grammatical errors too. Overall, writing is not coherent and not up to the mark. Try to join the paragraphs with joining lines (Lines 83 to 93 and jumping to completely different in next paragraph 95 to 102)

Comments on the Quality of English Language

Line 57: "decades one of the most productive areas" should be "decades, one of the most productive areas."

Line 58: "knows an exponentially growth" should be "knows exponential growth."

Line 73: "Harvesting of leaves is carried out before complete greening." - Ambiguous. Consider rephrasing for clarity, such as "Harvesting of leaves should be done before they are fully green."

Line 83: "Based on its reach phenolic" should be "Based on its rich phenolic."

Line 84-85: "agents in different models [8, 9], antihypertensive agents [10]" - Inconsistent comma usage in lists. Consider revising for consistency.

Line 119: "and a microwave-assisted method, for which the" - Consider removing the comma after "method."

Line 129: "For phytosynthesis, equal volumes" - The comma after "phytosynthesis" may not be necessary.

Line 157: "follows: 2-100% B at 30 °C for 60 minutes at an elution flow rate of 1 ml/min" - Consider clarifying "B" for readers unfamiliar with chromatography.

Line 173-174: "X-ray diffraction (XRD) analyses were performed using a 9 kW Rigaku SmartLab diffractometer" - It might be clearer to say "XRD analyses were performed with a 9 kW Rigaku SmartLab diffractometer."

Line 226: "Following the obtaining" could be more directly stated as "Following the extraction."

Round 2

Reviewer 1 Report

Comments and Suggestions for Authors

1. Absorption spectra of plant extracts without nanoparticles are just what is needed to evaluate the effect of nanoparticles on their appearance. Either the authors can take absorption spectra of solutions of their synthesized nanoparticles relative to solutions of extracts at a similar concentration, then perhaps the SPR band will be well resolved.

2. The absorption spectra of all objects should be taken at comparable concentrations. Figure 2c shows spectra of plant extracts taken apparently for very concentrated solutions (OD greater than 4). Matching concentrations is required.

3. Figure 3. Unknown compounds Mw and T are indicated in the legend

4. For the validity of interpreting the band at the noise level as a silver reflex, the authors should supplement the manuscript with a diffractogram of the isolated extracts they use for nanoparticle synthesis.

Author Response

The adsorption spectra of the crude extracts were inserted in the previous review round, in order to answer the reviewers’ comments. The adsorption spectra of the extracts at corresponding dilutions were present even from the original manuscript. In order to make it clearer, we also added the adsorption spectra of silver nitrate and added the dilution factor. Figure 2c (undiluted samples) was added to discuss the crude extracts spectra (and the information provided regarding their composition). 

We added the XRD spectra of the extracts and modified the legend of the figure 3. From Figure 3 it is now visible that the extracts do not contribute to the apparition of silver specific (111) plane.

Reviewer 2 Report

Comments and Suggestions for Authors

I have reviewed the corrected version of the manuscript and feel that further corrections to its content are necessary before it is suitable for publication.

1. The authors' response to comment 1 is unacceptable. The crystallite’s size and its accuracy of determination will give information, whether the nanoparticles are made of single crystallite or several crystallites. Since the size of crystallites is much smaller than the free path of electrons, their size significantly affects their properties. In most cases, TEM images provide information about the overall size without showing the number of crystallites.

Because they refuse to calculate the crystal size error. The authors should include a table with the parameters and their accuracy of determination, respectively according to the procedure described in the response to comment 2. As well as add a description of the procedure in the text of the manuscript.

2. In the newly added text in the discussion of IR spectroscopy results, the reference is missing in the following sentence in Page 14: "The results are in good concordance with available literature data (dddd)."

Author Response

In order to accurately calculate the crystal size error, the other specific diffraction peaks should be perfectly defined (this is not the case, as we can only define with confidence the (111) plane).

In order to provide a satisfactory response to the reviewer, all the discussion regarding the crystallite size was removed from the manuscript, and only particle size was maintained as a result of the study.

The reference absent from the revised manuscript was added.

Reviewer 3 Report

Comments and Suggestions for Authors

The authors made some corrections and added additional information that improved the quality of the article, but there are still some problems with the biological part of the paper.

The authors didn't present the data on cell viability and anti-inflammatory activity of crude horehound extracts. This information needs to be added.

Also, the LD50 values correspond to the semi-lethal dose in animals but not to the cytotoxicity. Since the MTT assay was used, the LD50 should be changed to IC50.

Author Response

Cell viability and anti-inflammatory activity of crude horehound extracts were added, and a comparison between the crude extracts and their corresponding nanoparticles was discussed in the text. 

The LD50 was changed to IC50.

Reviewer 4 Report

Comments and Suggestions for Authors

I didn't find any supplementary data as mentioned by the authors in their response of synthesis procedure. 

Comments on the Quality of English Language

I didn't find any supplementary data 

Author Response

We have tried to provide supplementary data on the synthesis procedure, by inserting in the text of the paper a exhaustive presentation of all the parameters of the phytosynthesis.

Round 3

Reviewer 1 Report

Comments and Suggestions for Authors

The authors have tried to take into account all the comments on the manuscript. A few minor comments still need to be corrected for a positive decision to accept the manuscript.

1. The x-axis captions of Figures 4d and 4h need to be made uniform.

2. The absorption spectra shown in Figure 2c are taken in the range of very high optical densities, which prevents observing the conditions of the Bouguer-Lambert- Beer law. It is required either to remove this figure from the manuscript or to reshoot it with a modified sample preparation protocol to obtain data in the range of optical density not exceeding 1.5.

(3) Correction of sentences on lines 385-388, 407-410 is required.

Comments on the Quality of English Language

Some sentences in the text are difficult to understand. A grammar check is required. 

Author Response

We thank the reviewer for the time spent in evaluating our manuscript and for the valuable remarks. We have corrected our manuscript according the reviewer's comments, all the corrections being presented using the track-changes function.

  1. The x-axis captions of Figures 4d and 4h were made uniform.
  2. Figure 2c was discarded from the manuscript.
  3. The indicated sentences were corrected. We also performed a grammar check and tried to make sentences more easily readable throughout the manuscript.

Reviewer 2 Report

Comments and Suggestions for Authors

I have no additional comments to the manuscript and I think it can be published in its present form.

Author Response

We thank the reviewer for the time spent in evaluating our manuscript and for the valuable remarks throughout the review process. 

Reviewer 3 Report

Comments and Suggestions for Authors

The authors have made all the necessary corrections and the article is ready for publication in Nanomaterials.

Author Response

(The authors gave the same response as above.)

Reviewer 4 Report

Comments and Suggestions for Authors

Manuscript can be accepted now

Author Response

(The authors gave the same response as above.)
